# *Shigella* Mutant with Truncated O-Antigen as an Enteric Multi-Pathogen Vaccine Platform

**DOI:** 10.3390/vaccines13050506

**Published:** 2025-05-10

**Authors:** Jae-Ouk Kim, Harald Nothaft, Younghye Moon, Seonghun Jeong, Anthony R. Vortherms, Manki Song, Christine M. Szymanski, Jessica White, Richard Walker

**Affiliations:** 1Science Unit, International Vaccine Institute, Seoul 08826, Republic of Koreaseonghun.jeong@ivi.int (S.J.); mksong@ivi.int (M.S.); 2VaxAlta Inc., Department of Medical Microbiology and Immunology, University of Alberta, Edmonton, AB T6G 2E1, Canada; nothaft@ualberta.ca (H.N.); cszymans@ualberta.ca (C.M.S.); 3Walter Reed Army Institute for Research, Silver Spring, MD 20910, USA; anthony.r.vortherms.ctr@health.mil; 4Complex Carbohydrate Research Center, University of Georgia, Athens, GA 30602, USA; 5PATH, Seattle, WA 98103, USA; jawhite@path.org; 6PATH, Washington, DC 20001, USA; rwalker@path.org

**Keywords:** *Shigella*, *Shigella* truncated mutant, fermentor culture, *Campylobacter jejuni* N-glycan heptasaccharide, dmLT adjuvant, combination enteric vaccines

## Abstract

**Background/Objectives**: Rising antibiotic resistance underscores the urgent need for effective vaccines against shigellosis. Our previous research identified the *Shigella flexneri* 2a truncated mutant (STM), a *wzy* gene knock-out strain cultivated in shake-flasks, as a promising broadly protective *Shigella* vaccine candidate. Expanding on this finding, our current study explores the feasibility of transitioning to a fermentor-grown STM as a vaccine candidate for further clinical development. **Methods**: The STM and STM-Cj, engineered to express the conserved *Campylobacter jejuni* N-glycan antigen, were grown in animal-free media, inactivated with formalin, and evaluated for key antigen retention and immunogenicity in mice. **Results**: The fermentor-grown STM exhibited significantly increased production yields and retained key antigens after inactivation. Immunization with the STM, particularly along with the double-mutant labile toxin (dmLT) adjuvant, induced robust immune responses to the conserved proteins IpaB, IpaC, and PSSP-1. Additionally, it provided protection against homologous and heterologous *Shigella* challenges in a mouse pulmonary model. The STM-Cj vaccine elicited antibody responses specific to the N-glycan while maintaining protective immune responses against *Shigella*. These findings underscore the potential of the fermentor-grown STM as a safe and immunogenic vaccine platform for combating shigellosis and possibly other gastrointestinal bacterial infections. **Conclusions**: Further process development to optimize growth and key antigen expression as well as expanded testing in additional animal models for the assessment of protection against *Shigella* and *Campylobacter* are needed to build on these encouraging initial results. Ultimately, clinical trials are essential to evaluate the efficacy and safety of STM-based vaccines in humans.

## 1. Introduction

Shigellosis, caused by *Shigella* spp., remains a significant public health concern, particularly in developing countries [1,2]. The disease is characterized by symptoms such as bloody diarrhea, fever, and abdominal cramps and can be fatal in young children and immunocompromised individuals, resulting in 42,000–94,000 deaths among children less than five years of age [3]. The emergence of antibiotic resistance and the absence of licensed vaccines against *Shigella* continue to hinder efforts to control shigellosis effectively. For these reasons, the WHO has prioritized the development of a *Shigella* vaccine [4,5]. However, this is complicated by the fact that *Shigella* comprises four species with over 50 serotypes [6]. Therefore, the search for broadly effective *Shigella* vaccines has required consideration of serotype diversity and prioritization of the most common ones [7]. This has led to most clinical trials using attenuated cells or O-specific polysaccharide conjugates representing the *S. flexneri*, 2a, 3a, and 6 as well as *S. sonnei* serotypes [8].

Although not as commonly studied as the other *Shigella* vaccine approaches, inactivated whole-cell *Shigella* vaccine candidates are immunogenic and protective in animals and have been demonstrated to be safe and immunogenic in phase 1 trials using formalin-inactivated *S. sonnei* and *S. flexneri* 2a, respectively [9,10,11,12]. Notably, the bacterial whole-cell approach has also been successful for inactivated cholera vaccines as well as ongoing phase 1 and 2 trials of the ETEC vaccine ETVAX [13,14]. The mucosal adjuvant double-mutant (R192G/L211A) heat-labile toxin (dmLT) of *Escherichia coli* [15] has been used as an adjuvant as well as an antitoxin antigen for the ETVAX vaccine and has shown very encouraging immunogenicity in infants in both Bangladesh and Zambia, suggesting that the use of dmLT may further enhance the effectiveness of inactivated whole-cell vaccines [13,16,17]. Therefore, the whole-cell approach for *Shigella* vaccines deserves renewed attention based on its potential and the success of similar strategies with other bacterial diseases.

In addition to sero-specific antigens, attention has begun to focus on the use of conserved protein antigens of *Shigella*, which may improve and broaden coverage with a lower number of individual vaccine components than needed for O-specific polysaccharide (OSP, also known as O-antigen)-based vaccine candidates. For example, conserved antigens of the type 3 secretion system, such as invasion plasmid antigens IpaB, IpaC, and IpaD as well as the pan-*Shigella* surface protein, PSSP-1, have been safe and protective against the most epidemiologically important *Shigella* serotypes when tested in animals [18,19,20,21,22,23,24]. We previously reported that we modified the inactivated whole-cell approach to better present the conserved surface antigens of *Shigella* by creating a *S. flexneri* 2a 2457T strain lacking the *wzy* gene [25]. The *wzy* gene encodes the O-antigen polymerase that is responsible for the polymerization of the O-antigen repeats in lipopolysaccharides. Its deletion results in a single repeat unit, increasing the exposure of conserved outer-membrane antigens [26]. Named the *Shigella* truncated mutant (STM), this can serve as an effective, broadly protective *Shigella* vaccine candidate in mice by enhancing the immune responses to conserved outer-membrane proteins such as PSSP-1 when cultured in a flask using an animal-based medium. The STM vaccine is characterized by one unit of OSP and is anticipated to increase the exposure of the conserved outer-membrane proteins to host immune systems. Earlier studies found generally improved immune responses and broad protection. These findings were improved when dmLT was used as a component of the vaccine [25].

In the present study, we examined the potential of a fermentor-grown formalin-inactivated STM grown in animal free media to provide the crucial data necessary before further advancing this vaccine candidate toward readiness for subsequent cGMP manufacturing and human clinical trials. Moreover, we investigated the STM’s capability as a carrier for heterologous antigens, specifically the *C. jejuni* N-glycan heptasaccharide [27,28], aiming to broaden its utility and impact in multi-pathogen vaccine development.

Our results demonstrated the successful transition to growth of the STM in fermentors with the retention of key *Shigella* antigens even after prolonged storage. The STM vaccine prepared in this manner induced immune responses to key antigens and protection against *Shigella*, particularly when co-administered with dmLT in mice. The potential of the STM to serve as a vector for expressing a key conserved N-glycan antigen from another foodborne gastroenteritis pathogen, *C. jejuni* [27], was also demonstrated, further extending its applicability to combat multiple pathogens. Considering that *Campylobacter jejuni* causes tens of millions of diarrheal cases each year and is a leading contributor to foodborne illness globally, its inclusion in vaccine development efforts is of major public health importance [29,30,31,32].

## 2. Materials and Methods

### 2.1. Bacterial Strains

*S. flexneri* serotype *2a* strain 2457T [33], serotype 3a, serotype 6 [34], *S. sonnei* 53G [35], STM [25], and STM-Cj were used in this study.

### 2.2. Preparation of Bacterial Strains for Challenge Studies

*S. flexneri* 2a strain 2457T, serotypes 3a, 6, and *S. sonnei* 53G were prepared as described previously [25]. A frozen aliquot of bacteria was thawed, streaked on a Bacto tryptic soy (BD, Sparks, MD, USA) agar plate containing Congo Red (Serva, Heidelberg, Germany),and incubated at 37 °C overnight. One Congo Red-stained colony was picked from the plate and cultured in Bacto tryptic soy broth at 37 °C and subcultured until reaching 0.5 OD at 600 nm (=2 × 10^8^ cfu/mL). The bacteria were subsequently diluted to optimal concentrations with PBS. We confirmed the colony-forming unit (cfu) value by serial dilution of each bacterial challenge solution in PBS and spreading appropriate dilutions onto agar plates containing Congo Red.

### 2.3. Preparation of the STM in Flask and Fermentor Culture

#### 2.3.1. Flask Growth

The *Shigella* truncated mutant (STM) was generated by deleting the *wzy* gene from the *S. flexneri* 2a 2457T strain by λ red recombineering [36,37] and prepared by shake-flask culture, following the same procedure outlined in the previous section for other *Shigella* strains. STM was inactivated by treatment with 0.13% formalin (Sigma, Steinheim, Germany) in PBS on a shaker for 2 h at a controlled room temperature of 22–23 °C (RT).

#### 2.3.2. Fermentor Growth

Instead of the medium used in shake-flask culture, an animal-free medium, APS Super Broth (BD Difco, Franklin Lakes, NJ, USA) was used for all work leading to production in the fermentor. The 10-L fermentor was inoculated with 100 mL of an overnight culture and incubated with at 37 °C, with impeller agitation at 200 rpm and air flow of 5 L/min until an OD of 2.5 was reached, and the cells were collected by centrifugation and then suspended in 1 L of PBS with 0.2% formalin. The inactivation was allowed to proceed overnight in an incubator at 25 °C under gentle shaking. After incubation, the cells were washed three times with PBS to remove residual formalin and then suspended in PBS to the desired concentration. Then, 100 µL of a pre-inactivation aliquot was plated onto 3 TSA plates and incubated at 37 °C for 5 days to check for any growth. No growth was observed. A pre-inactivation of the culture was plated to determine the concentration of colony forming units. Flask cultures were used in this study on the day of their production and inactivation. For one set of mouse experiments involving either antibody analysis or challenge following three-dose immunization, the inactivated cultures were stored at 4 °C and used up to the third dose. In contrast, fermentor-grown vaccine material was stored at 4 °C and utilized successfully in animal studies beginning approximately 6 months after production.

### 2.4. Expression of the C. jejuni N-Glycan on the Surface of the STM

For the expression of the *C. jejuni* N-glycan heptasaccharide (Cj-N-glycan) in the STM vaccine platform, the minimal biosynthetic operon from *C. jejuni* that included *gne* (UDP-GlcNAc/Glc 4-epimerase), *pglK* (flippase), and *pglHIJIA* (Cj-N-glycan GTases) was inserted into a vector suitable for integration into the chromosome of the STM strain. To do so, first, the kanamycin cassette replacing *wzy* was removed from the STM following the method of Datsenko and Wanner [36], resulting in strain STM Δ*wzy*. Next, *wzy* up- and down-stream sequences were amplified from STM gDNA with oligonucleotide combinations wzy-XhoI-F (5′-AATACTCGAGAAATTGAATATATGAAAAAACATATTTTGG-3′), wzy-BamHI-R (5′-TTATTGG-ATCCCATATTCGTAAGGTGATG-3′), wzy-BamHI-F (5′-AAGGATCCTAAAGGATG-TTAAAAATAGGGAAGTTATTGACC-3′), and wzy-SacI-R (5′-CCTCAGAGCTCCCATA-CTTCATACGTAATTCTAAAGCATG-3′). After treatment with the respective restriction enzymes, purified PCR products were inserted (3-arm ligation) into plasmid pBluescript KS digested with SacI and XhoI, generating a single BamHI site in between the *wzy* up- and down-stream fragments. Next, a DNA fragment containing the minimal Cj-*pgl* locus that carried a kanamycin cassette flanked by FRT sites between *pglK* and *pglH* was prepared as described previously [38] and inserted into linearized (BamHI digested) plasmid pBluescript KS(Δ*wzy*). The integration construct obtained after the SacI/KpnI digestion of the resulting plasmid pBluescript KS(*wzy*::*pgl*) was gel-purified and integrated into the chromosome of STM Δ*wzy* by λ red recombineering [36]. After removal of the kanamycin cassette, integration at the correct chromosomal location (the minimal *pgl* locus replacing *wzy*) was verified by PCR, with oligonucleotides hybridizing within the *pgl* locus and outside of the recombination event. One correct candidate that transcribed the *pgl* genes in the same orientation as the original *wzy* gene was named STM-Cj and used for further analyses.

### 2.5. Characterization of STM and STM-Cj by Western Blot

A 3-fold serial dilution of whole cells (5 × 10^7^, 1.66 × 10^7^, and 5.55 × 10^6^ cfu) of bacteria in PBS per lane were used for SDS-PAGE. Freshly prepared shake-flask-cultured STM whole cells were also included with the same cfu numbers as described above. Recombinant IpaB, IpaC, and PSSP-1 proteins (15 ng) were used as the positive control for each 4–20% SDS gel. Monoclonal antibodies 2F1A344 (1:1000 dilution, WRAIR), clone 7-3 (1:1000 dilution, IVI), and 2C12 (1:5000 dilution, IVI) were used as primary antibodies to detect IpaB, IpaC, and PSSP-1/IcsP, respectively, in western blot analysis. Horseradish peroxidase (HRP)-conjugated goat anti-mouse IgG (1:5000) was used as a secondary antibody. Western blot detection was performed using an Amersham™ Imager 680 (Cytiva, Marlborough, MA, USA). The intensity of each protein band was quantified by selecting the corresponding area and calculating the integrated density after background subtraction using ImageQuant TL software (Cytiva, Marlborough, MA, USA). Expression of the Cj-N-glycan-lipid A fusion was verified with Cj-N-glycan-specific antiserum (R1) after separation of proteinase-K-treated whole-cell lysates on 4–20% SDS gels, as previously described [38].

### 2.6. Flow Cytometry Analysis of Surface Exposure of PSSP-1 on Whole Cells

The same amounts of formalin-inactivated *S. flexneri* 2a 2457T, STM, and STM-Cj whole cells (1 × 10^7^ cfu) were incubated in dilutions of PSSP-1-specific polyclonal mouse sera [22] at 4 °C for 1 h. After washing 3 times in PBS, goat anti-mouse IgG-RPE (Southern Biotech, Birmingham, AL, USA) was added. After washing in PBS, cells were analyzed with a flow cytometry instrument (BD LSR II, BD Bioscience, San Jose, CA, USA). Naïve mouse serum was used a control.

### 2.7. Intranasal Immunizations and Challenges with Shigella

Female BALB/c mice, 6 weeks old, received inactivated whole cells (1 × 10^8^ cfu) of shake-flask-cultured *S. flexneri* 2a 2457T, fermentor culture STM, and STM-Cj in 40 µL of PBS by the intranasal route, 3 times, in 2-week intervals, under anesthesia. The STM and STM-Cj were administered alone or with 5 µg of dmLT produced by PATH [15]. Ketamine hydrochloride (Yuhan Co., Ltd., Seoul, Republic of Korea) (0.1 mg/g of body weight) and xylazine hydrochloride (Rompun; Bayer Korea, Seoul, Republic of Korea) (12.5 µg/g of body weight) were given via the intraperitoneal route for anesthesia. Mice were kept under specific-pathogen-free conditions at the IVI, and all animal experiments were performed with approval of the IVI Institutional Animal Care and Use Committee (protocol No. 2022-002). On day 13 after the last immunization, serum was collected from immunized mice, and, on the 14th day, the mice were intranasally challenged with live wild-type *S. flexneri* 2a 2457T (1 × 10^7^ and 5 × 10^7^ cfu), *S. flexneri* 3a (1 × 10^7^ and 5 × 10^7^ cfu), and *S. sonnei* 53G (5 × 10^6^ and 1 × 10^7^ cfu). Body weight and survival of mice were monitored daily for 14 days.

### 2.8. Enzyme-Linked Immunosorbent Assay

*Shigella*-specific serum IgG levels against recombinant protein, IpaB, IpaC [39], and IcsP [40]; *S. flexneri* 2a LPS and OSP; and *C. jejuni* N-glycan was measured by ELISA as described previously [25,38]. Briefly, 96-well plates (Nunc, Roskilde, Denmark) were coated with 200 ng/well of IpaB, IpaC, or PSSP-1, or with 500 ng/well BSA-Cj-N-glycan conjugate [41] in 100 μL of PBS at 4 °C overnight, while *S. flexneri* 2a LPS (200 ng/well) and OSP (100 ng/well) were diluted in citric acid and coated. After blocking with blocking buffer (1% BSA in PBS or 5% skim milk in 100 μL of PBS-T in the case of BSA-Cj-N-glycan), serial dilutions of sera or BAL fluids in blocking buffer were incubated for 2 h at RT. Then, HRP-conjugated goat anti-mouse IgG (1:5000, Southern Biotech, Birmingham, AL, USA) or (for the detection of N-glycan-specific antibodies) alkaline phosphatase (AP)-conjugated goat anti-mouse IgG (1:2500, Santa Cruz Biotechnology, Dallas, TX, USA) was incubated for 1 h at RT. After final washing, for the HRP-conjugated secondary antibodies, peroxidase substrate (TMB; Moss, Pasadena, MD, USA) was added to each well for 10–15 min, and 0.5 N HCl was added for stopping the reaction. For AP-conjugated secondary antibodies, detection was performed using pNPP as a substrate following the instructions of the manufacturer (Thermo Fisher, Waltham, MA, USA) at 405 nm. In both cases, the OD was measured in an ELISA reader (Molecular Devices, Sunnyvale, CA, USA). The antibody titer is expressed as the reciprocal log2 titer of dilution showing 0.2 absorbance at 405 nm for BSA-Cj-N-glycan and 450 nm for the other antigens.

### 2.9. Statistical Analysis

Data are expressed as mean ± standard deviation (SD). Statistical significance between the individual groups was analyzed using the unpaired Student’s *t* test. A log rank (Mantel–Cox) test was used for comparing survival rates after challenge. All analyses were conducted using Prism 10 (GraphPad, San Diego, CA, USA).

## 3. Results

### 3.1. Growth of STM in Fermentor Compared to Flask

The STM was cultivated in an animal-free medium using a 10-*L* fermentor until reaching an OD of 0.8. Fermentor culturing the STM resulted in a >2-log increase in production yield (2-fold by cfu per mL, >2-fold by bacterial particle count per mL) compared to the STM grown in a flask at a scale 300–500 mL.

### 3.2. Expression of Key Shigella Antigens in STM Grown by Fermentation with Animal-Free Medium

Key antigens of *Shigella*, including IpaB, IpaC, IcsP and LPS, were detected in a fermentor-grown batch of the STM by western blot analysis. Since IcsP is an outer-membrane protease, it was expected to be surface-exposed in its full-length form. PSSP-1 represents the C-terminal half of IcsP and was used in this study as a recombinant antigen. To assess the antigenic expression profile, the STM grown in a fermentor, using animal-component-free medium, was compared with the freshly grown STM, both of which were formalin-inactivated. Even after 6 and 9 months’ storage at 4 °C, the fermentor-grown STM vaccine maintained detectable levels of type III secretion system proteins IpaB, IpaC, and IcsP, a parental molecule of PSSP-1, as detected by western blot analysis. The expression levels of IpaB were comparable between the fermentor-grown and flask-grown STM, while lower levels of both IpaC and PSSP-1 were observed in the fermentor-grown STM (Figure 1). In summary, the fermentor-grown STM exhibited stability at 4 °C for up to 9 months, retaining detectable levels of key *Shigella* antigens, including IpaB, IpaC, and IcsP.

### 3.3. Immune Responses Induced by Fermentor-Cultured STM

When the inactivated fermentor lot of the STM, stored at 4 °C for approximately 6 months, was administered intranasally, it induced a similar level of IgG against IpaB and PSSP-1 but a lower level of IgG against IpaC compared to the freshly prepared inactivated wild-type *S. flexneri* 2a 2457T. The addition of dmLT to the STM induced potent IgG responses to IpaB and increased the IgG level to IpaC to a level similar to that of wild-type-alone immunization. However, the addition of dmLT to STM did not influence the IgG responses to PSSP-1. Despite the addition of dmLT, weaker responses to LPS and OSP were observed in the STM compared to those of the wild-type *S. flexneri* 2a 2457T, which was consistent with the *wzy* mutation leading to only one unit of OSP (Figure 2).

The dmLT-adjuvanted STM (STM + dmLT) provided higher protection to mice than STM alone against both the homologous *S. flexneri* 2a 2457T and the heterologous *S. flexneri* 3a J17B and *S. sonnei* 53G strains (*p* < 0.05 between PBS and STM + dmLT groups in all challenge experiments) (Figure 3A,C,D). Additionally, STM alone still exhibited significant protection compared to PBS (*p* < 0.05) against both the high dose of *S. flexneri* 2a 2457T as well as the low-dose challenges with the *S. flexneri* 3a J17B and *S. sonnei* 53G strains (Figure 3C,D). While there was no statistically significant difference, STM + dmLT showed greater protection than wild-type *S. flexneri* 2a against high-dose homologous challenge. The mice immunized with STM + dmLT showed faster recovery of body weight regardless of homologous challenge dose compared to those immunized with the wild-type parental strain (Figure 3B). Overall, our findings confirm that the fermentor-cultured STM, stored for 6 months, induced antibody responses to type III secretion system proteins and PSSP-1 as well as effectively protected mice when administered with dmLT.

### 3.4. Characterization of STM-Cj

The potential of the STM to serve as a carrier or vector for delivering other heterologous antigens was assessed (Figure 4). The STM, which expresses the *C. jejuni* N-glycan antigen on the surface, was constructed and designated STM-Cj. The expression of the STM structure, as well as the Cj-N-glycan linked to lipid A of the STM, was investigated by western blotting (Figure 4A). These studies demonstrated the successful integration of the *C. jejuni* biosynthetic locus, resulting in the detection of both the Cj-N-glycan and the truncated STM LPS structure linked to lipid A. Subsequently, we examined the impact of *C. jejuni* N-glycan antigen expression on the ability of the STM to express key *Shigella* antigens, including IpaB, IpaC, and IcsP, at levels comparable to those expressed in the absence of the Cj-N-glycan. We observed that STM-Cj retained IpaB, IpaC, and IcsP. However, STM-Cj expressed a similar level of IpaB but lower levels of IpaC and IcsP compared to its parent STM grown in a flask, as determined by western blot analysis (Figure 4B). Additionally, STM-Cj showed a similar level of IcsP exposure on the whole-cell surface to that of the STM cultivated in a fermentor (23.3% vs. 26.5%), whereas the wild-type parental *S. flexneri* 2a 2457T displayed only 5.5% [42] (Figure 4C).

### 3.5. Evaluation of STM-Cj Immunogenicity in Mice Against Shigella and Campylobacter Antigens

Mice were used to evaluate whether the antigens expressed on the STM-Cj were sufficient to induce immune responses. We measured the IgG levels against IpaB, IpaC, PSSP-1, LPS, and OSP (*S. flexneri* 2a) as well as Cj-N-glycan in the immunized serum by ELISA. STM-Cj induced IgG responses to IpaB and PSSP-1, with levels similar to those of the wild-type *S. flexneri* 2a. Additionally, the levels were elevated when administered with dmLT (Figure 5). However, the IpaC-specific IgG levels induced by STM-Cj in this set of experiments were even higher than those of the wild-type *S. flexneri* 2a 2457 regardless of the presence of dmLT. Additionally, STM-Cj induced lower levels of LPS and OSP (*S. flexneri* 2a)-specific IgG responses than those of the wild-type immunization, regardless of whether dmLT was added to STM-Cj. Also, STM-Cj elicited Cj-N-glycan-specific IgG responses in mice, but there was no enhanced effect when co-administered with dmLT (Figure 6). These results suggest that STM-Cj retains the ability to induce immune responses to both key *Shigella* protein antigens and Cj-N-glycan.

### 3.6. Protection Against Shigella Following STM-Cj Immunization in Mice

The capacity of STM-Cj to confer protection against *Shigella* challenges was evaluated (Figure 6). The STM-Cj construct demonstrated cross-serotype protection against *S. flexneri* 2a 2457T and *S. sonnei* 53G in the mouse lung challenge model, as was seen with the STM alone. In these experiments, when both the STM and STM-Cj were administered with the dmLT adjuvant, they induced comparable levels of homologous protection against lethal pneumonia following both high- and low-dose challenge (80 to 90% protective efficacy) (Figure 6A, upper panels). Additionally, both the STM and STM-Cj vaccine candidates plus dmLT provided comparable levels of heterologous protection against the low-dose challenge with the 53G strain of *S. sonnei* (ranging from 50 to 60 percent). However, the high-dose challenge with the *S. sonnei* 53G strain was lethal for several mice in the *S. sonnei*-immunized control group, and more mice died in other groups as well, resulting in an absence of observed protective effects for either the STM or STM-Cj (Figure 6A, lower panels). In addition, immunization with the dmLT-adjuvanted STM or the STM-Cj construct protected against weight loss following both the high- and low-dose challenges with *S. flexneri* 2a strain 2457T. The weight loss patterns in mice given the high-dose challenge with the 2457T strain of *S. flexneri* 2a are shown in Figure 6B, upper panels. The prevention of weight loss in mice given the low dose of *S. flexneri* 2a 2457T was similar in the STM- and STM-Cj-immunized mice, but in *S. sonnei* -challenged mice, protection against weight loss was less pronounced (Figure 6B lower panels).

## 4. Discussion

In this study, we successfully transitioned the preparation of an STM vaccine from animal media in shake flasks to growth in an animal-free medium in a fermentor to lay the foundation for the advanced process development necessary for the production of lots for vaccine clinical trials and eventually to a scale practical for commercial development. In addition, we demonstrated its capacity to serve as a vector for delivering heterologous antigens, specifically, the conserved N-glycan heptasaccharide of *Campylobacter*.

The fermentor-grown STM retains key *Shigella* antigens, including IpaB, IpaC, and PSSP-1, even after prolonged storage at 4 °C for up to 9 months. This stability is essential for ensuring the potency and efficacy of the vaccine candidate during storage and transportation, thus facilitating its clinical development. Additionally, the demonstrated stability of the STM supports its feasibility for large-scale production and distribution, addressing the significant challenges in vaccine manufacturing and deployment. However, despite the overall success of the fermentor-grown STM, it is important to note that the expression levels of IpaC and PSSP-1 were lower compared to the STM freshly cultivated in a flask, while the expression level of IpaB remained unchanged under both culture conditions. This suggests that further optimization of fermentor culture conditions may be needed to improve the expression level of these key proteins, consequently enhancing the vaccine’s potency to induce immune responses [43]. Given that the STM samples were inactivated with formalin prior to long-term storage, the protein expression profile was expected to reflect the state at the time of inactivation. Thus, the differences observed are more likely attributed to the production method rather than storage. Nonetheless, the potential effects of prolonged storage on protein integrity cannot be completely ruled out.

Our study highlighted the robust immune responses in mice to IpaB and IpaC induced by the fermentor-cultured STM, particularly when administered with dmLT adjuvant. The antibody responses to the type III secretion system were induced effectively, indicating the potential of the STM to elicit broadly protective immunity against *Shigella* infections associated with diverse serotypes. Interestingly, the IgG level to PSSP-1 was not elevated in the presence of dmLT in our experiments, and dmLT did not influence the levels of LPS and OSP. Due to resource limitations, we did not examine the IgA responses in the bronchoalveolar lavage fluid and focused on IgG levels in the serum. However, our earlier studies with a flask-grown STM indicated that the STM + dmLT group exhibited higher levels of IgA against conserved *Shigella* proteins, including IpaB, IpaC, and PSSP-1, compared to the group administered the STM alone, although this trend was not statistically significant. STM + dmLT immunizations also conferred protection to mice against both homologous and heterologous challenges with *S. flexneri* 3a, and *S. sonnei*, resulting in over 57% survival regardless of challenge dose.

Although some protection was seen in the STM cross-challenge experiments in mice with *S. sonnei*, there was a trend toward less protection than that observed with homologous *S. sonnei* challenge. This observation could be attributed to factors such as the unique capsular polysaccharide structure exclusively found in *S. sonnei*, which renders the bacterium less invasive due to the shielding effect on T3SS [44]. Consequently, the immune responses elicited by the STM, derived from *S. flexneri* 2a, might not adequately target T3SS and other outer-membrane proteins like PSSP-1 for defending against *S. sonnei*. This suggests that the breadth of coverage by the *S. flexneri* 2a component of the STM vaccine would need to be augmented with inclusion of an *S. sonnei* whole-cell component.

We hypothesized that the STM with a single O-antigen repeat attached to the *S. flexneri* lipid A core could also act as a carrier for other oligosaccharide antigens and not obstruct their presentation. In this study, we were able to demonstrate, using western blotting, the successful attachment of the *C. jejuni* N-glycan heptasaccharide at the lipid A core of the STM. The expression of two different capping oligosaccharides on the lipid A core raised the question whether the expression of the heterologous *Campylobacter* antigen was sufficient to induce a robust immune response and whether the immune response to *S. flexneri* antigens (particularly LPS) would be altered. This was determined by immunization studies comparing the STM-Cj delivered with or without dmLT in mice following intranasal administration. Not only did STM-Cj induce an N-glycan IgG response (Figure 5), the response to *S. flexneri* LPS was actually two-fold higher than when the *S. flexneri* LPS was capping the lipid A core alone (Figure 2). Although N-glycan titers were slightly higher when STM-Cj was combined with dmLT, the difference was not significant. In addition, the IgG titers against the N-glycan in this experiment showed individual variability, with 1–2 animals showing undetectable antibody responses (similar to that observed for OSP in Figure 2), whereas most titers against the *Shigella* antigens were consistently higher. This is in contract to the N-glycan responses observed using *E. coli* as the antigen carrier where the N-glycan in previous studies was engineered to cap all available lipid A cores. This indicates that modifications of N-glycan versus *S. flexneri* LPS expression levels, STM-Cj dose, or vaccine regimen modifications may be necessary to further enhance antibody development against the N-glycan in this construct. Once these critical factors are assessed/addressed in future vaccine process development studies, the STM-Cj construct will be ready to move onto more in-depth studies further evaluating immunogenicity and, more importantly, protective efficacy in animal models.

The N-glycan antigens produced by *C. jejuni* play a pivotal role in host interactions, particularly by facilitating adhesion to intestinal epithelial cells [45]. This suggests that inducing immune responses specific to N-glycan could provide protection against *C. jejuni* infections. Antibodies raised against the N-glycan antigen in mouse studies were also shown to be opsonizing in in vitro studies, facilitating the killing of wild-type *C. jejuni* [38]. Multiple studies in chickens have shown promise for N-glycan-based vaccines, which could reduce colonization by *C. jejuni* in this natural host by several logs [27].

STM-Cj retained the ability to induce immune responses to key *Shigella* protein antigens while eliciting specific responses against Cj-N-glycan. The presence of STM-Cj N-glycan did not mask PSSP-1 on the surface of whole cells of the STM, as shown by flow cytometric analysis. Additionally, STM-Cj induced a higher antibody level against IpaC in the absence of dmLT compared to the wild-type parental strain. However, unlike with IpaB and PSSP-1, the IpaC-specific IgG level did not increase even in the presence of dmLT. We will further examine the antibody levels during the optimization of culture conditions. Based on the effect of STM-Cj on the survival of immunized mice, it can be concluded that STM-Cj still retained the ability to protect against *Shigella* challenges. Taken together, the immune responses against *Shigella* and *Campylobacter* indicate that STM-Cj is a promising way to add a conserved *Campylobacter* antigen component to a multi-pathogen vaccine, a concept that warrants further development given the growing interest in combination vaccines.

## 5. Conclusions

This report confirms and extends previous work by developing the STM as an effective *Shigella* vaccine. It underscores that the STM is scalable and can act as a safe, immunogenic, and broad-coverage vaccine against *Shigella*. Its potential value to protect against enteric diseases was enhanced by its construction as a vector for a conserved and protective *Campylobacter* antigen, the N-glycan. With further development, the STM-Cj construct could be an effective vaccine against two enteric pathogens for which there is currently no licensed vaccine. Moreover, if administered in conjunction with other orally administered whole-cell vaccines such as oral cholera vaccines and/or the ETVAX vaccine for ETEC, the impact of the STM-Cj vaccine could be even further magnified. An advantage of orally administered bacterial whole-cell vaccines, such as those described here, is that separate licensed products have the potential to be administered in conjunction in a modular fashion as multiple sips rather than parenteral injections, thereby facilitating their development, evaluation, and use, should they be shown to be effective. Our previously reported data in the murine respiratory challenge model showed that the STM provided protection against the four clinically relevant serotypes and species of *Shigella* [25]. These earlier findings were indicated here by showing protection against *S. flexneri* 2a and *S. sonnei* after growth and preparation were further developed. Additional challenge studies in another animal model such as the guinea pig Sereny test would be helpful to further confirm the broadness of protection that may be seen in future clinical studies. Whether used as a stand-alone or modular strategy, the data presented here certainly warrant the further process development and formulation of this vaccine candidate given its potential benefit to human populations.

## Figures and Tables

**Figure 1 vaccines-13-00506-f001:**
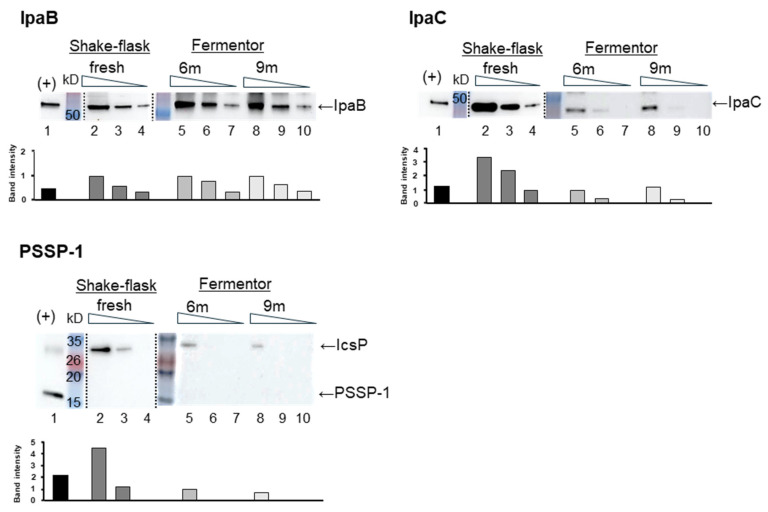
Comparison of expression levels of typical *Shigella* proteins, IpaB, IpaC, and IcsP, in STM depending on culture conditions. Whole cells of STM grown in fermentor were formalin-inactivated, aliquoted in PBS, boiled with SDS buffer, and stored at −20 °C until use. For SDS-PAGE, bacterial suspensions of 5 × 10^7^, 1.66 × 10^,^ and 5.55 × 10^6^ cfu per lane were loaded (lanes 5–10), alongside freshly prepared shake-flask-cultured STM in lanes 2–4 with the equivalent numbers. Recombinant IpaB, IpaC, and PSSP-1 proteins (15 ng each) served as positive controls in lane 1 for each respective protein. Primary antibodies in western blot were 2F1A344 for IpaB (**upper left**), clone #7-3 for IpaC (**upper right**), and 2C12 for PSSP-1/IcsP (**lower** panel) detection. Band intensity ratios of each protein are expressed as bar graph beneath each western blot panel.

**Figure 2 vaccines-13-00506-f002:**
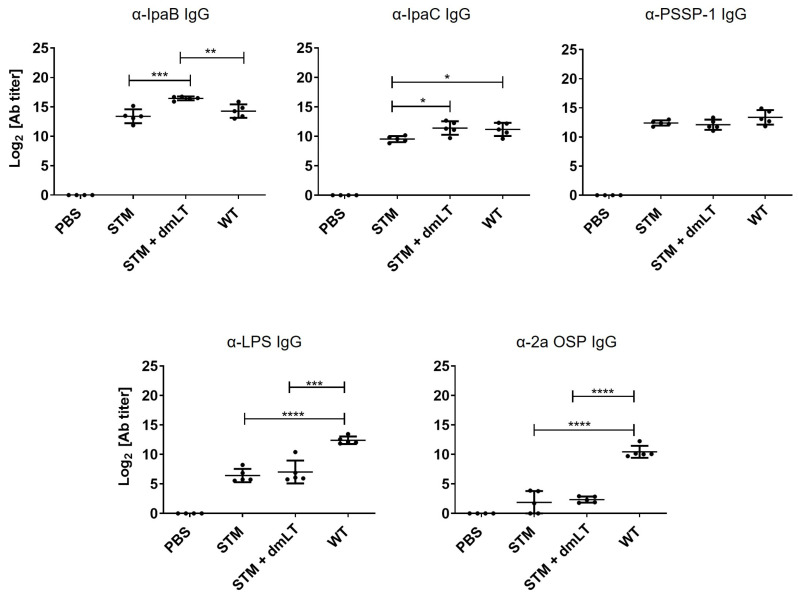
Immune responses following immunizations with STM. Mice were intranasally immunized three times in two-week intervals with inactivated whole cells (1 × 10^8^ cfu) of fermentor-grown STM with or without dmLT (5 μg) and *S. flexneri* 2a 2457T (WT). Seven days after 3rd immunization, serum was collected from individual mice, and IgG titers were determined by ELISA. Data are expressed as mean ± SD. N = 5 per group, * *p* < 0.05, ** *p* < 0.01, *** *p* < 0.001, and **** *p* < 0.0001.

**Figure 3 vaccines-13-00506-f003:**
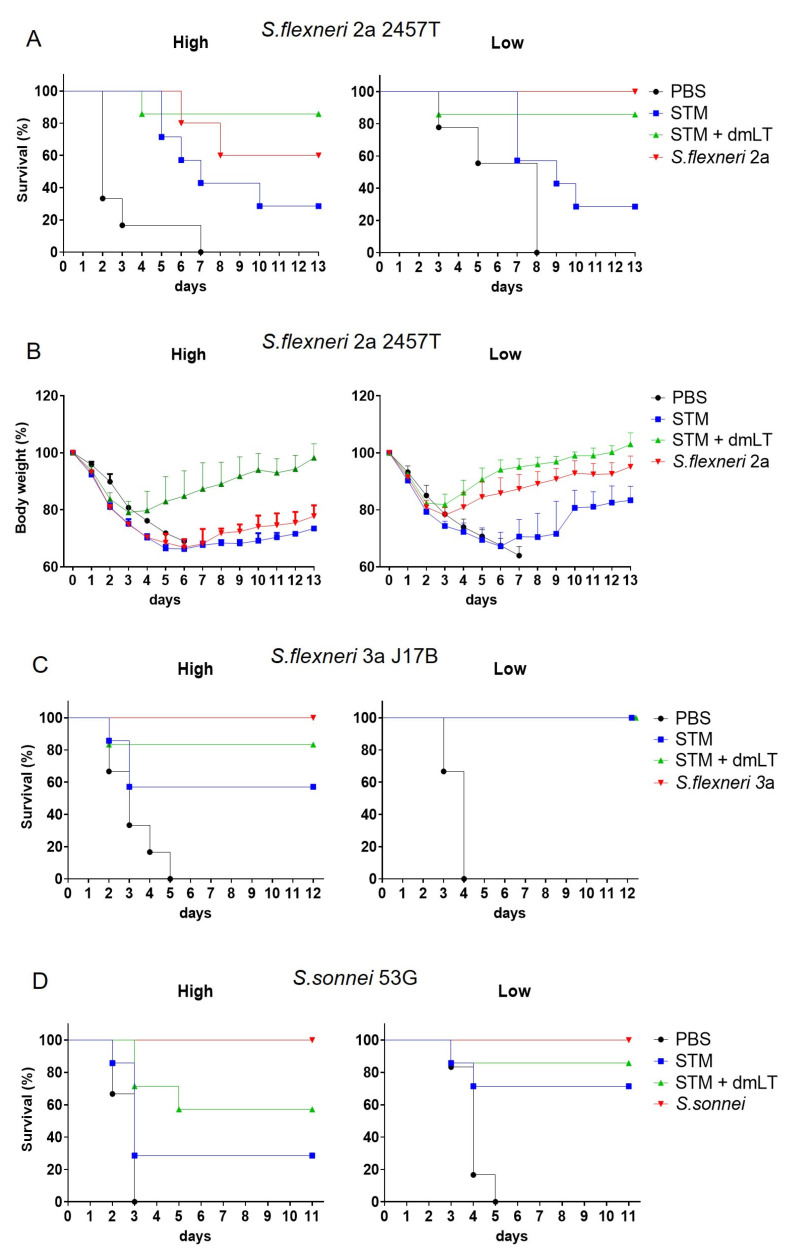
Protective immune responses following STM immunizations. One week after the third immunization, including inactivated whole cells of wild-type strain as positive control, mice were i.n. challenged with wild-type strains of *S. flexneri* 2a 2457T (panels **A** and **B**), *S. flexneri* 3a J17B (panel **C**), and *S. sonnei 53G* (panel **D**), each with high (**left** panels) and low (**right** panels) doses. Survival of mice was monitored daily, and there were seven mice in each group. After homologous challenge with *S. flexneri* 2a 2457T, body weight was monitored daily.

**Figure 4 vaccines-13-00506-f004:**
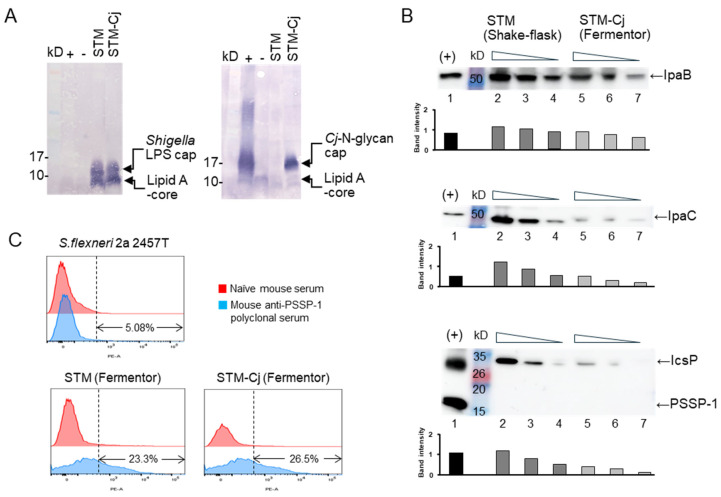
Construction of STM-Cj and comparison of expression levels of typical *Shigella* proteins with STM. (**A**) Construction of STM-Cj: **left** panel, western blot with *Shigella* LPS-specific antiserum; **right** panel, western blot with Cj-N-glycan-specific antiserum (R1) probed against proteinase K-digested whole-cell lysates separated by 5–20% PAGE. (+) *E. coli* K12 expressing Cj-*pgl* operon on plasmid pACYC184 (−) *E. coli* K12 (pACYC184); STM, *Shigella* truncated mutant; STM-Cj, *Shigella* truncated mutant expressing the Cj-*pgl* operon from the chromosome. (**B**) Inactivated whole cells of STM-Cj grown in fermentor were aliquoted in PBS, boiled with SDS buffer, and stored −20 °C until use. For SDS-PAGE, bacterial suspensions of 5 × 10^7^, 1.66 × 10^7^, and 5.55 × 10^6^ cfu per lane were loaded (lanes 5–7), alongside freshly prepared shake-flask-cultured STM in lanes 2–4 with equivalent numbers. Recombinant IpaB, IpaC, and PSSP-1 proteins (15 ng each) served as positive controls in lane 1 for each respective protein. Western blots for IpaB (**upper**), IpaC (**middle**), and PSSP-1/IcsP (**lower**) are shown. Band intensity ratios of each protein are expressed as bar graph beneath each western blot panel. (**C**) IcsP surface exposure was analyzed on *S. flexneri* 2a 2457T grown in flask (**upper**), as well as STM (**lower left**) and STM-Cj (**lower right**), both grown in fermentor. Inactivated whole bacterial cells were stained with PSSP-1-specific mouse serum and PE-conjugated goat anti-mouse polyclonal IgG (blue histogram). Naïve mouse serum was used as negative control (red histogram).

**Figure 5 vaccines-13-00506-f005:**
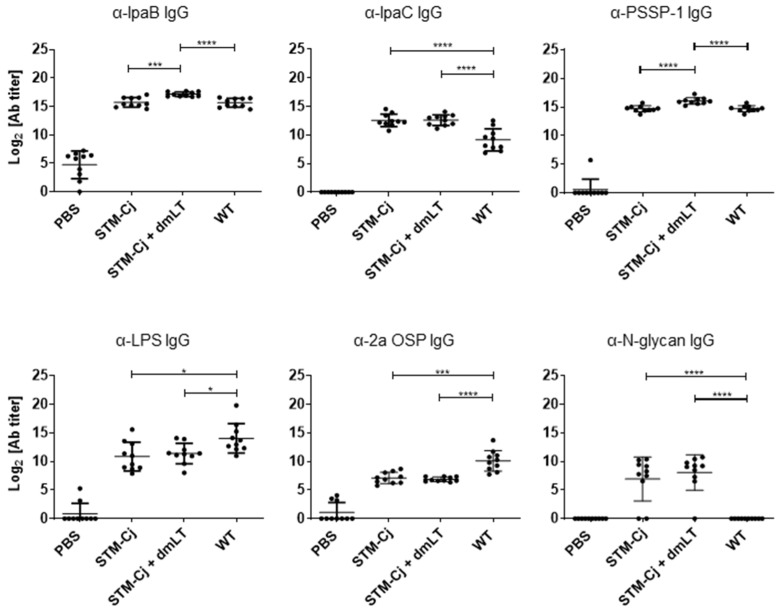
Antibody responses induced by STM-Cj. Mice were immunized intranasally 3 times in 2-week intervals with formalin-inactivated whole cells (1 × 10^8^ cfu) of shake-flask-cultured STM and fermentor-cultured STM-Cj with or without dmLT (5 μg) and *S. flexneri* 2a 2457T (WT). Nine days after 3rd immunization, serum was collected from individual mice, and IgG titers were determined by ELISA. **Upper left**, anti-IpaB IgG; **upper middle**, anti-IpaC IgG; **upper right**, anti-PSSP-1 IgG; **lower left**, anti-*S. flexneri* 2a LPS IgG; **lower middle**, anti-*S. flexneri* 2a OSP IgG; **lower right**, anti-Cj-N-glycan IgG. Data are expressed as mean ± SD. N=9–10 per group; * *p* < 0.05, *** *p* < 0.001, and **** *p* < 0.0001.

**Figure 6 vaccines-13-00506-f006:**
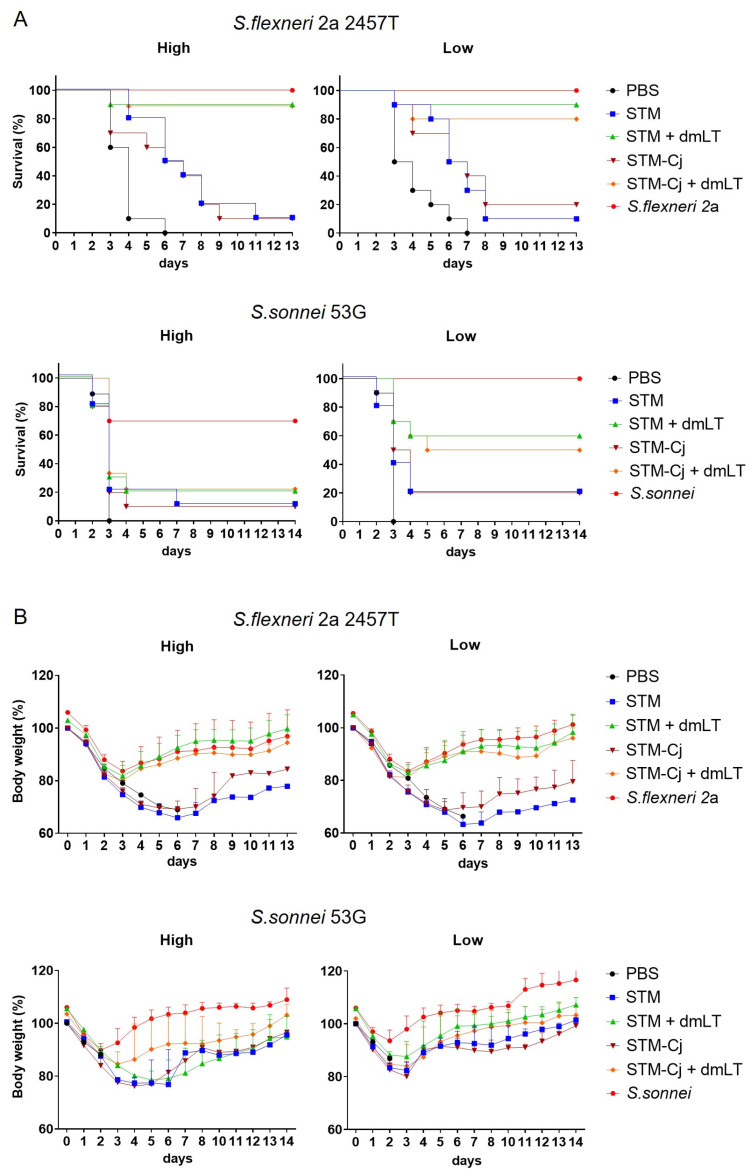
Survival and body weight recovery of mice immunized with STM-Cj ± dmLT against homologous and heterologous *Shigella* challenges. Mice were intranasally immunized 3 times in 2-week intervals with inactivated whole cells of 1 × 10^8^ cfu of shake-flask-cultured STM ± dmLT (5 μg), fermentor-cultured STM-Cj ± dmLT (5 μg), and wild-type *S. flexneri* 2a 2457T or *S. sonnei* 53G as positive control for each challenge experiment. Two weeks after the third immunization, mice were intranasally challenged with wild-type *S. flexneri* 2a 2457T (**upper** panels) and *S. sonnei* 53G (**lower** panels) with high (**left** panels) and low (**right** panels) doses. Survival (**A**) and body weights (**B**) of mice were monitored daily. N = 9–10 per group.

## Data Availability

The raw data supporting the conclusions of this article will be made available by the authors on request.

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
