# Peer review of "Shigella Mutant with Truncated O-Antigen as an Enteric Multi-Pathogen Vaccine Platform"

_vaccines, 2025, doi:10.3390/vaccines13050506_

Round 1
Reviewer 1 Report
Comments and Suggestions for Authors
The paper by Kim et al investigates the efficacy of Shigella flexneri 2a truncated mutatnt as a potential vaccine carrier to help prime immune responses to provide protection aginst Shigellosis. They have incorporated a variety of conserved proteins to determine if they offer cross protection frm the vaccine strain and other natural pathogens.
The paper is well written and clear to read and data is well presented in a clear format. Just a few minor points.
Introduction:
No mention of what is the wzy gene in the background or of its importance to Shigella
What was the source of the monoclonal antibodies 2F1A344 clone 7-3 and 2C12 , were used as primary antibodies to detect IpaB, IpaC and PSSP-1/IcsP, respectively, in western blot analysis.
Methods:
Should have a section that describes the various bacterial strains used in the experiments
e.g. S. flexneri 2a 2457T, STM and STM-Cj, S. sonnei 53G
Have shown that the ST mutant when delivered with mucosal adjuvant can protect against low or high dose pathogen challenge, but what is not clear is how broad the protection may be in protecting against all 50 serovars of Shigella strains? Show cross protection against 2 is encouraging but that is not to say the vaccine strain is going to be able to offer broader protection.
As a proof-of-concept study this is solid – but a lot more work needed to get this to be able to assess broad cross strain protection and this has been acknowledged by the authors in the discussion.
Author Response
Reponses to Reviewer 1
The paper by Kim et al investigates the efficacy of Shigella flexneri 2a truncated mutatnt as a potential vaccine carrier to help prime immune responses to provide protection against Shigellosis. They have incorporated a variety of conserved proteins to determine if they offer cross protection from the vaccine strain and other natural pathogens.
The paper is well written and clear to read and data is well presented in a clear format. Just a few minor points.
We thank Reviewer 1 for the insightful and constructive comments. We have addressed each point as follows:
Introduction:
Comment 1: No mention of what is the wzy gene in the background or of its importance to Shigella
Response 1: We have added a description of the wzy gene in the Introduction to clarify its function and relevance: "The wzy gene encodes the O-antigen polymerase that is responsible for the polymerization of O-antigen repeats in lipopolysaccharides. Its deletion results in a single repeat unit, increasing the exposure of conserved outer membrane antigens." (Line 70-73)
Comment 2: What was the source of the monoclonal antibodies 2F1A344 clone 7-3 and 2C12 , were used as primary antibodies to detect IpaB, IpaC and PSSP-1/IcsP, respectively, in western blot analysis.
Response 2: The source of antibodies (2F1A344, clone 7-3, and 2C12) is now described in the Methods section. 2F1A344 was provided by WRIAR and clone 7-3 and 2C12 were developed by IVI. (Section 2.5, Line 167-168)
Methods:
Comment 3: Should have a section that describes the various bacterial strains used in the experiments
e.g. S. flexneri 2a 2457T, STM and STM-Cj, S. sonnei 53G
Response 3: We added a new section under Methods titled "2.1. Bacterial Strains" that includes S. flexneri serotype 2a strain 2457T, serotype 3a, serotype 6, S. sonnei 53G, STM, and STM-Cj (Line 97-99).
Comment 4: Have shown that the ST mutant when delivered with mucosal adjuvant can protect against low or high dose pathogen challenge, but what is not clear is how broad the protection may be in protecting against all 50 serovars of Shigella strains? Show cross protection against 2 is encouraging but that is not to say the vaccine strain is going to be able to offer broader protection.
As a proof-of-concept study this is solid – but a lot more work needed to get this to be able to assess broad cross strain protection and this has been acknowledged by the authors in the discussion.
Response 4: We have demonstrated protection against a representative set of Shigella strains (S. felxenri 2a, S. flexneri 3a, S. flexneri 6 and S. sonnei). These 4 are by far the major causes of human disease. The next goal is to see if similar levels of protection against the 4 major Shigella strains can be achieved in another animal model prior to moving to human trials. The following text was added to the discussion to clarify this point:
Our previously reported data in the murine respiratory challenge model showed that the STM would provide protection against the four clinically relevant serotypes and species of Shigella. These earlier findings were indicated here by showing protection against S. flexneri 2a and S. sonnei after growth and preparation of the STM vaccine candidate were further developed. Additional challenge studies in another animal model such as the guinea pig Sereny test would be helpful to further confirm the broadness of protection that may be seen in future clinical studies (Line 491-497).

Reviewer 2 Report
Comments and Suggestions for Authors
The authors have previously developed the Shigella truncated mutant (STM), wher eby the wzy polymerase of Shigella flexneri 2a was inactivated to prevent polymerization and generate a truncated O-antigen. This was designed to improve exposure of immunogenic, conserved cell-surface protein antigens of Shigella. This publication adapted the existing method of STM production to be suitable for cGMP manufacture using non-animal based media and fermenter culture rather than shake flask. This method still enabled expression of IpaC, IpaB and IcsP antigens, generated an immune response and was provided a level of protection against S. flexneri 2a, 3a and S. sonnei in a murine infection model. STM was shown to be a viable expression platform for the conserved Campylobacter jejuni N-glycan, while retaining protective efficacy against S. flexneri 2a infection, offering the potential to develop a multi-pathogen diarrheal vaccine (although the protective effect of the conserved C. jejuni pgl locus is not yet proven). dmlT was also demonstrated as a key adjuvant for boosting the immune response and protective efficacy of the vaccine candidate. One challenge of Shigella vaccine development is the focus on the O-antigen as the major immunogen to generate protective efficacy, as this is serotype and species specific. This paper focuses on the enhanced exposure of conserved cell-surface protein antigens to generate a O-antigen independent immune response. Developing an immune response to these offers the potential for a pan-Shigella vaccine, avoiding the serotype-specific restrictions of an O-antigen based approach. The paper includes updating the STM production method towards cGMP manufacture. A better understanding of this process is important to support development of early stage vaccine candidates into large scale production.
Overall, this is an interesting paper demonstrating an alternative vaccine approach for Shigella, where there remains no licensed human vaccine. It is an innovative paper that should appeal to a wide readership of Vaccines.
Suggested changes
- Line 66- PSSP1 should be PSSP-1
- Line 113- what dilution was the fermenter inoculated with and what were the incubation condition
- Line 119- a pre-inactivation “aliquot” of the culture rather than pre-inactivation culture would improve clarity
- Line 121-flask cultures used for one set of experiments- clarify here which experiments these were
- Section 2.3- it would be useful to specify which genes or at least the first and last gene of the pgl locus used for glycan expression as it is difficult to extract this from the referenced paper
- Line 192- BSA-Cj conjugate used for coating ELISA plates- where did this come from? Production not described in the methods.
- Statistical significance reported in results but no details on statistical tests used in methods or results
- Why were fermenter and shake flask production of STM and protein expression never directly compared?
- Figure 1- compared protein expression on STM from shake flask cultures to STM from fermenter that had been stored for 6-9 months, can’t determine whether lower protein expression is due to culture method or storage.
- How is band intensity calculated for protein expression in Figures 1 and 4?
- Figure 4 A in legend the Shigella blot is describe as the right panel but I think this is the left and vice versa?
- In the figure legend, specify this is Cj N-glycan specific anti-serum
- Line 278- “when not vectoring” is a bit unclear- expressing may be better?
- It is unclear how IcsP and PSSP-1 are linked, I believe PSSP-1 is a fragment of IcsP and this is briefly mention in line 220 but there is constant swapping between the two terms throughout the text which can be confusing. It would be useful to explain what each are more fully and what would be expected on the surface- IcsP or PSSP-1?
- Fig 4B- again comparing fresh shake flask to frozen fermenter STM, one is fresh and one frozen- could storage or culture methods also be impacting protein expression levels? This hasn’t been determined.
Reviewer 3 Report
Comments and Suggestions for Authors
This manuscript give some interesting information, however it also need major revision before it can be considered for acceptance in this journal. Some detailed suggestion was shown below:
- Title, should be specific to gastrointestinal bacterial pathogen;
- Introduction, should introduce the role of the pathogens used in this study in the gastrointestinal bacterial disease;
- Materials and Methods, only STM was used in this study, it seems that this wild type strain S. flexneri 2a 2457T also should be included as a control, if not, please explain why?
- it is better to add some experiments or include some data about the underlying mechanism of STM vaccine;
- Results, some data can be presented as Tables, now tables;
- For figures, each gel land and column should be marked using number and explain in notes;
- This permission for the animal study should be provided or included in this manuscript.
